# A Pocket Guide to CCR5—Neurotropic Flavivirus Edition

**DOI:** 10.3390/v16010028

**Published:** 2023-12-23

**Authors:** Amit Garg, Jean K. Lim

**Affiliations:** Department of Microbiology, The Icahn School of Medicine at Mount Sinai, One Gustave L. Levy Place, Box 1124, New York, NY 10029, USA; amit.garg@mssm.edu

**Keywords:** chemokine receptor, CNS, leukocyte trafficking, genetic susceptibility

## Abstract

CCR5 is among the most studied chemokine receptors due to its profound significance in human health and disease. The notion that CCR5 is a functionally redundant receptor was challenged through the demonstration of its unique protective role in the context of West Nile virus in both mice and humans. In the nearly two decades since this initial discovery, numerous studies have investigated the role of CCR5 in the context of other medically important neurotropic flaviviruses, most of which appear to support a broad neuroprotective role for this receptor, although how CCR5 exerts its protective effect has been remarkably varied. In this review, we summarize the mechanisms by which CCR5 controls neurotropic flaviviruses, as well as results from human studies evaluating a genetic link to CCR5, and propose unexplored areas of research that are needed to unveil even more exciting roles for this important receptor.

## 1. Introduction

CCR5 belongs to a family of 7-transmembrane G-protein-coupled receptors that mediates leukocyte recruitment through the engagement of its ligands, CCL3, CCL4, and CCL5 [1,2]. CCR5 was believed to function redundantly with other closely related receptors due, in part, to the coexpression of multiple related chemokine receptors in specialized leukocyte subsets. The idea of functional redundancy among these receptors was further supported by the general failure of chemokine receptor antagonists to alter the clinical course of immune/inflammatory diseases [3]. To further this tenet at the genetic level, individuals lacking CCR5 due to the presence of two copies of a commonly found 32-base-pair deletion (*CCR5Δ32* homozygotes) were found to have essentially identical health and lifespan compared to CCR5-sufficient individuals. The loss of CCR5 is, in fact, a genetic asset, as it is linked with several health benefits, the most remarkable of which is resistance to HIV infection [4,5,6,7,8]. While CXCR4 can also serve as a coreceptor for HIV entry, CCR5 is essential for HIV transmission, which was most clearly demonstrated by the identification of *CCR5Δ32* homozygotes that are resistant to HIV infection [9,10,11]. However, the hypothesis that CCR5 is a dispensable gene was challenged by the finding that Ccr5-deficient mice, as well as *CCR5Δ32* homozygotes, are more susceptible to West Nile virus (WNV), a mosquito-transmitted neurotropic flavivirus [7]. In the nearly two decades following these initial studies, the role of CCR5 has been examined in the context of numerous other medically important neurotropic flaviviruses, including tick-borne flaviviruses, Japanese encephalitis virus (JEV), and Zika virus (ZIKV) [12,13,14]. Most of these studies collectively indicate that CCR5 is broadly neuroprotective, yet we are still uncovering novel mechanisms by which this receptor can offer protection against this group of viruses. Here, we discuss the studies that have evaluated Ccr5 in mice in the context of several neurotropic flaviviruses; review the genotype: phenotype studies that have tested for the frequency of *CCR5Δ32* homozygotes in cohorts of neurotropic flavivirus-infected individuals; and propose new areas of research and/or tools needed to further advance our understanding of CCR5.

## 2. CCR5 Guides Effector Leukocytes into the CNS

Flaviviruses are vector-borne RNA viruses and a major cause of neuroinfections globally. These viruses can emerge and re-emerge unexpectedly in human populations, causing meningitis, encephalitis, paralysis, and death, and those who survive are often left with long-term neurologic sequelae [15]. Neurotropic flaviviruses were not considered a significant cause of CNS infections in the US until 1999, when WNV was identified as the cause of an outbreak of encephalitis in the New York City area. WNV spread across the US unimpeded and has caused annual outbreaks ever since, with a total of 56,575 cases and 2776 deaths from 1999 to 2022 (www.cdc.gov, accessed on 11 October 2023). Today, WNV is a major cause of encephalitis and the predominant flavivirus that circulates in the US. Studies of immunocompetent wild-type (WT) mice showed that infection with WNV (NY99 strain) resulted in the development of encephalitis, with mice succumbing to infection 7–12 days post infection, with a mortality rate of 30–40% [16,17]. Evaluation of the CNS tissue showed a high level of induction of several inflammatory chemokines, such as Ccl2, Ccl3, Ccl4, Ccl5, and Cxcl10, in response to viral infection in the brain. The roles of these chemokines and their receptors have been systematically evaluated *in vivo*, including Ccr2 [18], Cxcr3 (and its ligand, Cxcl10) [17,19], and Cxcr4 [20,21]. Since three of the ligands for Ccr5 (Ccl3, Ccl4, and Ccl5) were induced in the CNS following WNV encephalitis, the role of this receptor was investigated in mice by comparing WNV pathogenesis in wild-type (WT) versus *Ccr5*^−/−^ mice. Infection in *Ccr5*^−/−^ mice was uniform fatal, with no mice surviving infection beyond eight days post infection, while the majority of the WNV-infected WT mice survived. Analysis of the CNS revealed an inability to control viral replication that was coupled with a global loss of leukocyte infiltration, with a >50% decrease in CD4^+^ T, CD8^+^ T, NK cells, and peripherally derived monocytes in the CNS compared to WNV-infected WT mice (Figure 1). Strikingly, no differences were observed in viral replication, cell activation, or cytokine induction outside of the CNS, suggesting that Ccr5′s primary role was to guide peripheral leukocytes into the barrier-protected WNV-infected brain tissue [16]. The protective role of Ccr5 was replicated in another study, although the phenotype was less dramatic, with 50% mortality observed in Ccr5-deficient mice infected with WNV (NY99) compared to 20% in WT mice [22]. The authors of this study showed that the loss of Ccr5 resulted in an earlier onset of disease compared to WT mice, which coincided with increased Ccr5 ligand induction and blood–brain barrier (BBB) permeability within the cortical region of the brain. Total leukocyte infiltration was significantly reduced in the absence of Ccr5, driven largely by the loss of CD4^+^ T cells, CD8^+^ T cells, and peripherally derived monocytes. These observations are supported by several studies that have demonstrated that Ccr5 promotes the recruitment and/or generation of memory CD8^+^ T cells in mice [23,24]. Interestingly, no differences in leukocyte infiltration were observed in other regions of the brain or the periphery, suggesting that CCR5 may control cell migration through the BBB with finer specificity than previously thought.

To evaluate the role of CCR5 in human WNV infections, several groups have probed the frequency of *CCR5Δ32* homozygous individuals, who naturally lack CCR5 expression, in cohorts of neurotropic flavivirus-infected individuals, allowing for genotype: phenotype association studies. An initial study evaluated the frequency of *CCR5Δ32* homozygotes among WNV-infected patients (*n* = 395) from two US states (Arizona and Colorado) who experienced severe West Nile fever or neuroinvasive disease compared to a large group of random US blood donors (*n* = 1318) who provided samples before 1999 and were therefore presumed to be WNV-uninfected. *CCR5Δ32* homozygosity was enriched from 1.0% in the control population to 4.3% in the symptomatic WNV patient population from both Arizona and Colorado (OR = 4.5; *p* < 0.0001), with the enrichment of *CCR5Δ32* being highest in Arizona individuals with fatal outcomes (OR = 13.2; *p* = 0.03) [25]. To further validate these findings using a geographically matched control population, a cohort of symptomatic WNV-seronegative patients from Arizona was also genotyped (*n* = 125), among whom the frequency of *CCR5Δ32* was found to be 0.7%. A followup study with two additional symptomatic WNV-infected cohorts sampled from Illinois and California also showed an enrichment for *CCR5Δ32* homozygotes (3.6%; OR = 3.7, *p* = 0.002) when compared to the same reference cohort of random blood donors utilized in the aforementioned study (*n* = 1318) [26]. A third study identified WNV-infected individuals through screening of blood donations between 2003 and 2008. This study was fundamentally different from the previous two, as the vast majority of the WNV-infected individuals enrolled were either asymptomatic or developed mild symptoms that did not result in a doctor’s visit or hospitalization. The frequency of *CCR5Δ32* homozygotes among asymptomatic WNV-infected individuals was 0% (*n* = 169), while those who developed one or more symptoms associated with WNV accounted for 3.2%, suggesting that all *CCR5Δ32* homozygotes that were infected with WNV developed some degree of symptomatology. These data also demonstrate that *CCR5Δ32* homozygosity is not a risk factor for infection with WNV (as is the case for the presence of CCR5 and HIV infection) but for the development of symptoms following WNV infection [27].

Two additional studies were conducted that did not show any association between *CCR5Δ32* homozygosity and WNV infection. The first study compared WNV-infected individuals who developed a neuroinvasive disease (meningitis or encephalitis; *n* = 821) to WNV-infected individuals who did not (*n* = 1233) and found no significant change in *CCR5Δ32* homozygote frequency (1.58% in neuroinvasive disease cases versus 1.05% in non-neuroinvasive cases; OR = 1.51) [28]. While this study did not include a WNV-uninfected population, the frequency of *CCR5Δ32* homozygosity was not higher than expected in a WNV-negative population. Another study also failed to find a significant association of *CCR5Δ32* homozygosity when comparing symptomatic to asymptomatic WNV-infected individuals (OR = 2.14, *p* > 0.05). However, when the *CCR5Δ32* homozygous frequencies of the symptomatic WNV-infected individuals and all WNV-infected individuals from their cohort were compared to that of the reference cohort of random blood donors used in previous studies (*n* = 1318) [26], a significant difference was observed, with an OR = 1.38 (*p* < 0.05) or 1.36 (*p* < 0.01), respectively) [29]. A recent meta-analysis was conducted that supports a role for CCR5 in preventing the development of severe WNV disease, with an overall OR of 1.29 (*p* = 0.005) [30]. It is unclear why some studies show a clear association with WNV disease severity while others do not, but these discrepancies may be due to the rigor of the control populations used, as some of the aforementioned studies utilized a reference cohort, while others utilized an asymptomatic WNV-infected cohort, both of which have strengths and weaknesses. Well-defined control populations matched geographically and ethnically are ideal for studies seeking to define a role for *CCR5Δ32*. Interestingly, no change in susceptibility to WNV is observed among *CCR5Δ32* heterozygotes, whose expression of CCR5 is reduced by ~30–50%. These data suggest that lower levels of CCR5 expression are sufficient to confer protection against WNV [31].

## 3. CCR5 Directs Regulatory T-Cell Migration to the CNS

Given that CCR5 appeared to offer protection in the context of WNV infection, the next major question was to understand whether these protective effects were relevant for other neurotropic flaviviruses. To address this, Ccr5 was next evaluated in a mouse model of Japanese encephalitis virus (JEV). JEV is a mosquito-borne flavivirus antigenically related to WNV and the predominant cause of viral encephalitis in Asia [32]. Over three billion individuals are at risk of infection due to endemic JEV transmission in 24 countries in the western Pacific and southeast Asia areas. According to the World Health Organization (WHO), China and India have suffered the greatest JEV burden, with 33,488 and 21,572 cases reported over the last 20 years, respectively (www.who.int, accessed on 11 July 2023). Like WNV, there is an immunocompetent mouse model to study JEV pathogenesis. Different chemokines and chemokine receptors have also been investigated in the context of JEV infection *in vivo*. One study showed that the loss of Ccr2 resulted in fewer Ly6c^hi^CD11b^+^ monocytes in the JEV-infected CNS and a survival advantage, while the loss of Ccl2 resulted in increased Ly6c^hi^CD11b^+^ monocytes and T cells (both CD4^+^ and CD8^+^) in the CNS, as well as increased mortality, suggesting fewer leukocytes in the CNS and, thus, less inflammation, resulting in a protective effect [33]. Likewise, examination of the Cxcr3 and its ligands, Cxcl10 and Cxcl4, also suggested that less inflammation in the CNS is associated with favorable outcomes, with either the blockade of Cxcl10 or Cxcr3, or Cxcl4 deficiency, resulting in increased viral clearance, less inflammation, and an overall protective effect [34,35,36].

In the context of Ccr5, one study found that infection with JEV (Beijing-1) resulted in 54% mortality in WT mice, which increased to 100% in Ccr5-deficient mice. Unlike WNV, the loss of Ccr5 in the context of JEV did not increase CNS viral loads but instead caused an early dysregulation in the production of cytokines and chemokines in the CNS, resulting in a surprising increase in neutrophils and, to a lesser extent, monocytes and CD8^+^ T cells. This was concomitant with decreased numbers of NK cells and regulatory CD4^+^ T cells (Tregs). Given that Tregs can express Ccr5 and are known to moderate inflammatory responses induced by pathogens, the authors hypothesized that Tregs may be critical in controlling the aberrant inflammation caused by the excessive influx of neutrophils and increased cytokine production. Adoptive transfer of Ccr5-sufficient Tregs but not Ccr5-deficient Tregs into JEV-infected *Ccr5*^−/−^ mice was able to restore survival to ~50% (similar the survival rate of WT mice) and ameliorate paralysis through production of the anti-inflammatory cytokine IL-10 and TGF-β. However, neutrophil and monocyte numbers remained elevated in the CNS of Ccr5-deficient mice reconstituted with Ccr5-expressing Tregs. This may have been due to the timing of the Treg cell transfer, which took place on day 3 post infection, a time point at which both neutrophils and monocytes had already entered the CNS [13]. Overall, these data show that Ccr5 provides neuroprotection in the context of JEV by promoting the infiltration of Tregs in the CNS, which can offset the inflammatory damage caused by an early and pathogenic presence of neutrophils and other innate immune cells (Figure 2).

The role of CCR5 was also evaluated in the context of Langat virus (LGTV) infection, a naturally attenuated member of the tick-borne encephalitis virus (TBEV) complex. Tick-borne flaviviruses are endemic throughout most of the northern hemisphere, causing neuroinvasive disease of varying severity, with mortality reported to be as high as 20–30% [37]. Infection with LGTV (TP21) in Ccr5-deficient mice resulted in an overall survival rate that was decreased (48% survival) compared to WT mice (90% survival). Reminiscent of observations in the CNS of mice infected with JEV, Ccr5-deficient mice infected with LGTV presented with decreased levels of T cells and NK cells in the CNS, along with a significant increase in neutrophil infiltration. The increased mortality observed in *Ccr5*^−/−^ mice was partially reversed by depleting neutrophils just prior to viral entry into the CNS or adoptive transfer of Ccr5-sufficient splenocytes into LGTV-infected *Ccr5*^−/−^ mice [12]. While the role of Tregs was not formally evaluated in this study, it appears that the protection offered by Ccr5 in the context of LGTV is similar to that of JEV. The role of Tregs in the CNS has also been shown to be critical during WNV encephalitis both in mice and humans [38,39,40,41], and the critical role of Tregs may be to inhibit neutrophil infiltration, a function that has recently emerged in various other inflammatory settings [42,43].

Testing the role of CCR5 in the context of human JEV infections has been challenging, since the homozygous frequency of *CCR5Δ32* is much lower in populations in which JEV circulates and causes disease, including in China and India. One study evaluated a cohort from northern India for *CCR5Δ32* frequency in a population of 183 JEV cases versus 361 geographically matched healthy controls [44]. No *CCR5Δ32* homozygotes were identified in either cases or controls, consistent with the expected homozygous frequency in India, which is estimated to be ~0.04% (1 in 2500) [45]. In China, *CCR5Δ32* homozygosity is expected to be even less frequent, estimated to be ~0.002% (1 in ~50,000) [45]. This contrasts with many European and northern European populations, where the *CCR5Δ32* homozygous frequency is between 1.0% (1 in 100) and 2.6% (1 in 36) [45]. Thus, testing the extent to which CCR5 is neuroprotective in the context of JEV infections will be difficult to address.

Because TBEV circulates primarily in European countries, association studies with *CCR5Δ32* and TBEV infections have been more insightful. The first study, conducted in a Lithuanian population, showed a significant enrichment for *CCR5Δ32* homozygotes (2.3%) in severely TBEV-infected individuals (*n* = 129), while the frequency in geographically matched negative controls (*n* = 134) and a group of TBEV-negative patients presenting with aseptic meningoencephalitis (*n* = 76) was 0% (*p* = 0.026) [46]. A second study found a similar correlation between *CCR5Δ32* homozygosity and severe TBEV cases (*n* = 349), with the *CCR5Δ32* frequency reported at 2.3% compared to the control population (*n* = 210), where the authors observed no *CCR5Δ32* homozygotes (*p* = 0.023) [47]. However, no association was found in a study in a Russian population, where the *CCR5Δ32* homozygotes were found in 2.9% of TBEV cases (*n* = 137) versus 1.5% among controls (*n* = 268; *p* > 0.05), although the frequency of *CCR5Δ32* homozygosity trended towards a pathogenic effect [48]. Interestingly, one study evaluated PBMCs and cells obtained from the cerebrospinal fluid of CCR5 WT or *CCR5Δ32* heterozygous individuals acutely infected with TBEV and found that while CCR5 expression was lower on the cells obtained from *CCR5Δ32* heterozygotes, this did not impact TBEV susceptibility or response [49]. Consistent with observations in WNV, these results further support the notion that haploinsufficiency for CCR5 does not increase susceptibility to TBEV.

## 4. NK and CD8^+^ T-Cell Effector Functions Are CCR5-Dependent

NK cells are cytotoxic lymphocytes critical to innate immunity and are functionally analogous to cytotoxic T cells (CTLs). A subset of both NK cells [50] and CD8^+^ [51] CTLs are known to express CCR5, and the role of these effector cells within the CNS has also been shown to be a critical part of CCR5′s arsenal. NK cells and CD8^+^ T cells function in the direct killing of virally- infected cells through the release of effector cytokines, such as IFNγ and TNFα, and through degranulation. Ccr5′s role during JEV was tested by another group utilizing the Nakayama strain [52]. Consistent with a JEV study conducted in mice utilizing the Beijing-1 strain [13], the loss of Ccr5 resulted in a significant increase in mortality (64% compared to 28% observed in WT mice). In addition to observing a delay in the influx of CD8^+^ T, CD4^+^ T, NK cells, and monocytes into the CNS, the authors evaluated the functional capacity of the effector cells. To do this, NK cells were isolated from the spleens of Ccr5-sufficient or Ccr5-deficient JEV-infected mice on day 4 post infection. They found that the NK cells isolated from the Ccr5-deficient mice were ~50% less effective at killing target cells *ex vivo* and produced less IFNγ compared to NK cells obtained from JEV-infected Ccr5-sufficient mice (Figure 3). Similarly, CD8^+^ CTLs isolated from JEV-infected Ccr5-deficient mice on days 4 and 7 post infection were less capable of producing IFNγ and/or TNFα following *ex vivo* stimulation with an immunodominant JEV-derived peptide or whole virus compared to WT mice. Collectively, these data suggest that the loss of Ccr5 not only reduces the number of leukocytes capable of reaching the CNS but that the NK and CTLs that do reach the brain are further crippled functionally, leaving the host suboptimally equipped to fight virally infected neurons within the CNS [52]. Several studies have shown a critical role of Ccr5 in guiding and enhancing interactions of T cells with dendritic cells, suggesting that T cells may not be sufficiently primed, which may, in part, explain this phenotype [24,53,54,55]. Once in the CNS, perforin-dependent granule exocytosis is a major pathway required for the successful clearance of numerous viruses, including flaviviruses [56,57]. Currently, it is unknown how the loss of Ccr5 on these effector cells might result in reduced functionality, and more work is needed to understand this functional defect.

## 5. CCR5 Promotes Viral Persistence in the Brain Endothelium

The most recent flavivirus to cause a large-scale epidemic is Zika virus (ZIKV), which first made headlines due to an outbreak in Micronesia in 2007, followed by its introduction into the western Hemisphere that started in Brazil and rapidly spread through South and Central America in 2015–2016 [58]. Similar to other neurotropic flaviviruses, ZIKV is vector-transmitted through a mosquito bite, but it is unique, as it can be vertically transmitted from a mother to a fetus [59]. Infection of pregnant individuals with ZIKV, irrespective of symptom development, can lead to microcephaly and other congenital malformations, as well as fetal loss, stillbirth, and preterm birth. Since the initial outbreak, there has been a global reduction in ZIKV cases, but this virus still circulates in at least 89 countries (www.who.int, accessed on 8 December 2022). Given that there are no vaccine or antiviral options and the potential for another large-scale outbreak, ZIKV still poses a significant global concern, particularly for people who are pregnant or planning to become pregnant. Another interesting feature of ZIKV is the sexual transmission that has been documented [60,61,62,63,64]. Furthermore, ZIKV infection in humans can lead to viral persistence in some tissues, such as the testes, for months following initial exposure, allowing the virus to be transmitted through seminal fluid long after the acute phase of the infection has passed [65,66]. It was previously shown that barrier endothelia, such as the blood–brain barrier (BBB) [14] or the blood–testes barrier (BTB) [67], can become persistently infected with ZIKV without triggering cytopathology, offering a reservoir and possible mechanism for viral persistence. Using an *in vitro* human brain microvascular endothelial cell (hBMEC) model to study persistent infection of the BBB, one study showed that the CCR5:CCL5 interaction was critical for the survival and persistence of ZIKV-infected hBMECs. Blocking the interaction of CCL5 with CCR5 on ZIKV-infected hBMECs knocked-out for either CCR5 or CCL5 resulted in a significant decrease in ZIKV infection (Figure 4) and a reduction in the cell viability of the hBMECs, suggesting that the CCR5:CCL5 interaction promotes the survival of the ZIKV-infected brain endothelium, allowing for persistent infection. Blockade of this interaction using CCR5-specific antagonist Maraviroc recapitulated this phenotype, resulting in the rapid clearance of ZIKV from the endothelial cells [14,68]. While these *in vitro* data suggest a pathogenic role of CCR5 in the context of ZIKV infection, it is unclear to what extent this mechanism would impact ZIKV infection *in vivo*. While CCR5 appears to be expressed on hBMECs *in vitro*, whether it is expressed on the BBB in a steady state or under inflammatory conditions *in vivo* has not been characterized. Furthermore, it is unclear whether the loss of CCR5 on the BBB would promote pathogenesis or protect against ZIKV in mice and humans. While CCR5 deficiency may abrogate ZIKV persistence in the long term, even a transient breach in the BBB caused by a loss of viability of the ZIKV-infected brain endothelium in the absence of CCR5 could be detrimental to the host in the short term. It is possible that CCR5 could play a protective role during the acute phase of infection while still promoting viral persistence in the BBB. To study the role of Ccr5 in the context of ZIKV infection *in vivo*, utilization of the recently generated humanized STAT2 knock-in mice would be the most powerful model, as these mice are immunocompetent [69].

To date, no studies have evaluated the association between *CCR5Δ32* and ZIKV infection. Similar to challenges faced with JEV, the frequency of *CCR5Δ32* homozygosity in Brazil and other countries in South and Central America is lower than in European populations [45,70]. Future studies must also keep in mind that the allele frequency for *CCR5Δ32* can vary widely, even within a country, such as Brazil, where the frequency of *CCR5Δ32* homozygotes ranges between 0.03% (~1 in 3300) and 0.86% (~1 in 116) depending on the region [70]. Thus, future cohort studies that seek to associate ZIKV outcomes with loss of CCR5 in flavivirus-endemic countries must carefully control for this.

## 6. CCR5-Mediated Leukocyte Infiltration Promotes Pathogenesis

While the loss of leukocyte infiltration into the CNS in the absence of Ccr5 often results in increased mortality, there are studies that have shown that fewer leukocytes in the CNS as a result of Ccr5 deficiency can be beneficial as well. In fact, Ccr5-deficient mice infected with Rocio virus, a neurotropic flavivirus in the JEV serocomplex that circulates in South America [71], displayed a greater survival rate (~40%) compared to WT mice, where infection was uniformly fatal [71]. The authors showed that brain viral loads between the WT and Ccr5-deficient mice were similar, but leukocyte infiltration was greatly reduced in the *Ccr5*^−/−^ brains. Likewise, another study showed that the mortality observed in mice intraperitoneally infected with JEV (NJ2008 strain) could be reversed through the administration of CCR5 antagonist Maraviroc (20% survival without Maraviroc to 70% survival in Maraviroc-treated mice) [72]. Similar to the studies involving Rocio virus, treatment with Maraviroc resulted in fewer leukocytes in the CNS. Together, these results suggest that Ccr5 guides leukocytes into the virally infected CNS, but the collateral damage from the cellular infiltration outweighs the benefit of viral clearance offered by their presence, leading to a better survival outcome in the absence of Ccr5. These studies suggest a detrimental role for Ccr5 in the context of some neurotropic flaviviruses, and more in-depth studies are needed to further confirm these results.

## 7. Future Directions

Most of studies conducted to date appear to align with a neuroprotective role for CCR5 that is enabled through numerous mechanisms that likely overlap *in vivo*, depending on the pathogen and model system used (Table 1). It remains unclear whether the protective role of CCR5 may also extend beyond the CNS to include other barrier-protected tissues, including the testes and the placenta-protected fetus. To test this, ZIKV would be the best candidate pathogen, as this virus is known to breach both the BTB to infect the seminiferous tubules and cross the placenta to infect the developing fetus, both of which occur in the context of ZIKV infection in humans and have also been reported in mice [73,74]. It is important to note that while most *in vivo* studies in mice appear to show a protective role for CCR5, there are several studies that have demonstrated that loss of Ccr5, and the concomitant loss of leukocyte infiltration into the CNS may improve survival by lowering the immunopathology. Furthermore, the impact of CCR5 in the context of flavivirus persistence is completely unknown; for this, a model of flavivirus persistence will need to be established.

Maraviroc is a weak antagonist against mouse Ccr5; this is not surprising, given the ~77% identity between mouse and human CCR5 at the protein level. While Maraviroc can block human CCR5 ligand binding with an IC_50_ of 1.9nM, this is dramatically increased to 3600nM when tested against the mouse receptor [75]. Another study showed that Maraviroc was 76 times less effective at inhibiting signaling through mouse Ccr5 compared to human CCR5 [76]. Given this drastic decrease in efficacy, most mouse studies utilize high levels of Maraviroc that should be sufficient to block mouse Ccr5 *in vivo*, but off-target effects of Maraviroc are completely unknown. Thus, the findings of studies that seek to understand the role of Ccr5 in mice utilizing Maraviroc should be validated utilizing another methodology if possible.

To fully understand the multitude of mechanisms by which CCR5 impacts inflammation and immunity in the context of any disease, the field needs several new reagents and tools. Currently, there is considerable confusion about the expression of Ccr5 on cells in tissues and through differentiation and development in mice in the steady state and during any immune process, all of which stems from the low specificity of the commercially available anti-mouse Ccr5 antibodies. Thus, a new suite of antibodies that are highly specific for mouse Ccr5 and able to not only identify cells expressing this receptor but also block chemokine binding/chemotaxis are necessary. The availability of a blocking antibody would also provide a more effective way to assess Ccr5 function in mice and allow researchers to circumvent the use of Maraviroc. To resolve some of the controversy regarding the cellular expression of Ccr5, a reporter mouse for Ccr1, Ccr2, Ccr3, and Ccr5 was recently described [77], which showed that Ccr5 was preferentially expressed in tissue resting macrophages and T cells. It has been immensely informative in assigning expression profiles in leukocytes in blood and tissues in the steady state and in some inflammatory conditions, but individual chemokine receptor reporter knock-in mice may be necessary to assign receptor-specific functions in the context of specific inflammatory or infection conditions. For these purposes, a Ccr5 reporter mouse, as well as a floxed Ccr5 transgenic mouse, would be incredibly useful. There is a multitude of questions that still need to be addressed regarding how CCR5 functions broadly in health and disease. In addition to the generation of new tools to study Ccr5, investigation into additional flaviviruses and assessment of the role of Ccr5 in barrier-protected tissues besides the CNS, such as the testes, as well as in the context of pregnancy, will help unlock even more exciting roles for this multifunctional receptor.

## Figures and Tables

**Figure 1 viruses-16-00028-f001:**
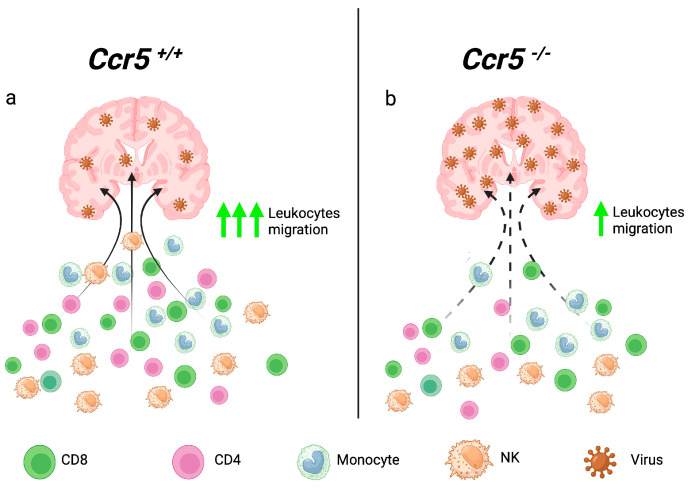
Ccr5 controls leukocytes trafficking into the CNS. Following entry and replication within the CNS, neurotropic flaviviruses, such as WNV, trigger the recruitment of CD4^+^ T cells, CD8^+^ T cells, NK cells, and peripherally derived monocytes in wild-type (*Ccr5^+/+^*) mice (**a**). However, in the absence of Ccr5 (*Ccr5*^−/−^), these cell subsets are unable to migrate into the CNS, eventually leading to uncontrolled virus replication and mortality. Dotted arrows represent reduced cell migration into the infected brain (**b**). CNS: central nervous system; WNV: West Nile virus; NK: natural killer.

**Figure 2 viruses-16-00028-f002:**
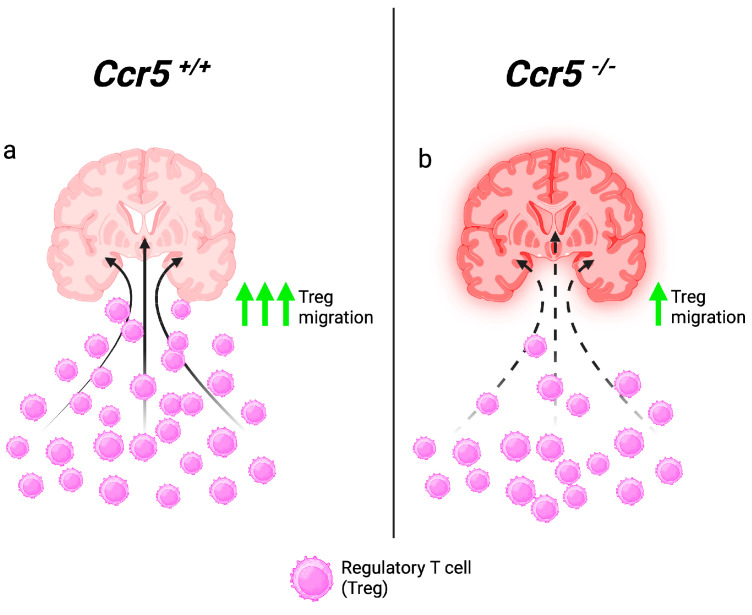
Ccr5-expressing regulatory T cells mitigate CNS inflammation. After entry into the CNS, neurotropic flaviviruses such as JEV and LGTV trigger inflammation mediated by both innate immune cells and cytokines in the CNS that is mitigated by the presence of regulatory T cells (Tregs) guided into the CNS through Ccr5 expression in wild-type (*Ccr5^+/+^*) mice. Solid arrows represent the direction of Tregs into the infected site (**a**). In the absence of Ccr5, Treg migration into the CNS is impaired, leading to uncontrolled inflammation of the brain tissue (represented by the red shadow). Dotted arrows represent reduced migration of Tregs into the infected brain (**b**). CNS: central nervous system; JEV: Japanese encephalitis virus.

**Figure 3 viruses-16-00028-f003:**
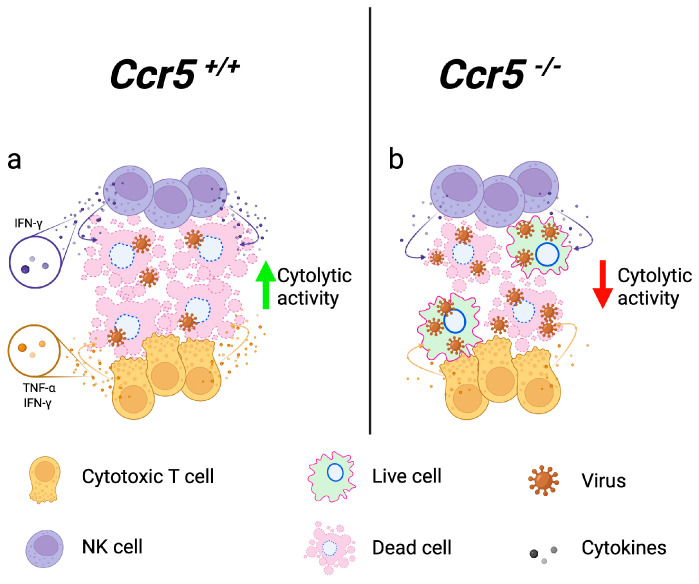
Optimal effector functions of NK and CD8^+^ T cells require Ccr5. NK cells and effector CD8^+^ T cells isolated from flavivirus-infected mice were tested for effective killing of target cells *ex vivo* and production of IFNγ and/or TNFα after *ex vivo* stimulation with immunodominant JEV-derived peptide or whole virus. Compared to wild-type *Ccr5^+/+^* mice (**a**), NK cells and CD8^+^ cells isolated from *Ccr5*^−/−^ mice were functionally impaired, leading to increased viral replication in the CNS (**b**). NK: natural killer; JEV: Japanese encephalitis virus; IFN: interferon; TNF: tumor necrosis factor.

**Figure 4 viruses-16-00028-f004:**
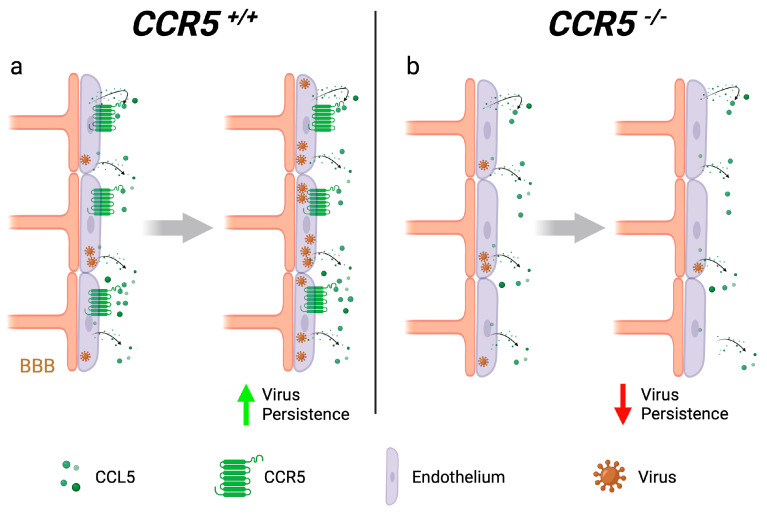
CCR5 promotes persistence of flaviviruses in the brain endothelium. Neurotropic flaviviruses, including ZIKV, can infect the brain endothelium *in vitro*. Signaling events induced by CCL5 binding to its receptor, CCR5, lead to persistent, long-term infection of the infected endothelium (**a**). Blocking the CCL5:CCR5 interaction, as would be the case in *CCR5Δ32* homozygosity, triggers cytopathology of the virus-infected endothelial cell that ultimately promotes viral clearance (**b**). ZIKV: Zika virus.

**Table 1 viruses-16-00028-t001:** In vivo role of CCR5 during neurotropic flavivirus infections.

Neurotropic Flavivirus	Protective	Pathogenic	No Association
West Nile virus	Glass et al. [16] Durrant et al. [22] * Glass et al. [25] * Lim et al. [26] * Lim et al. [27] * Cahill et al. [30]		* Loeb et al. [28] * Bigham et al. [29]
Japanese encephalitis virus	Kim et al. [13] Larena et al. [52]	Liu et al. [72]	* Deval et al. [44]
Langat virus	Michlmayr et al. [12]		
Tick-borne encephalitis virus	* Kindberg et al. [46] * Mickiene et al. [47]		* Barkhash et al. [48]
Rocio virus		Chavez et al. [71]	

* Human cohort studies.

## Data Availability

Not applicable.

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
