# Peer review of "A Pocket Guide to CCR5—Neurotropic Flavivirus Edition"

_viruses, 2023, doi:10.3390/v16010028_

Round 1

Reviewer 1 Report

Comments and Suggestions for Authors

This is a timely review of the role of CCR5 in flavivirus infection.  The scope includes human studies and mouse models of infection.  The overall presentation is balanced to give a good sense of solid advances versus preliminary results.  Dr. Lim is an international expert on both CCR5 and flaviviruses having performed several of the key studies in the field.

Specific criticisms:

1.       A Table summarizing the biochemical and biological properties of CCR5, including the cell types expressing it would be useful. 

2.       A Table summarizing the human and mouse studies of CCR5 for all the human flaviviruses would be useful.  Several flaviviruses are not mentioned in the review including yellow fever virus and dengue virus (PMID: 25939314, PMID: 24607451).  Those mentioned at the end only should be worked into earlier sections of the review.  It’s important to convey that the role of CCR5 in flavivirus infection is not settled science as these discrepant studies make clear.

3.       The introduction assigns four words and one reference for the role of CCR5 in HIV infection.  Although the review is focused on flaviviruses, it is important to state that the role of CCR5 in viral infection is most clearly established for HIV, where it works as a direct entry factor/coreceptor with CD4 essential for R5 HIV infection of CD4+ T cells and macrophages.  And that CCR5delta32, which is central to many of the studies reviewed here, was discovered as part of a search to understand the essential role of CCR5 in AIDS pathogenesis.  Compared to this clearly established role with known mechanisms, understanding of CCR5’s apparently sub-essential role in flavivirus infection is still a work in progress.

4.       The lack of agreement among the gene association studies for WNV and CCR5D32 needs more comment regarding the limitation of such studies.  Were any of these gold standard genome wide association studies?  Were any controlled for population structure?  Were they designed based on cohorts of convenience?  The characterization of the control populations is very important in interpreting these results.

5.       Although the focus is on CCR5, there would certainly be interest among readers about other chemokines/chemokine receptors that have been shown to be as or more important in these infections.  The only  one cited is a CXCL10 study.

6.       The mechanistic aspects of CCR5’s role still appears to be weak.  I think there are floxed Ccr5 mice to define the role of CCR5 on specific leukocyte subtypes in flavivirus infection. 

7.       The final reference is to a paper describing CCR5 reporter mice, but the precise cell types expressing CCR5 and related receptors described in the paper are not detailed in the review.  Readers will want to know what cell types express CCR5 ligands and CCR5 in the brain during flavivirus infection.  And whether CCR5 deficiency results in a defect in production of virus antigen-specific CD4 and/or CD8+ T cells.

8.       The published results with maraviroc are of questionable significance as the authors suggest.  Have any of the cited studies been performed with human CCR5 knockin mice?  What doses of the drug were used and are those doses able to achieve drug levels in the blood consistent with blocking mouse Ccr5?

9.    The WHO references need years and page numbers.  All the references should be proofread since there appear to be grammatical errors in some of the titles.  Also the text needs careful proofreading to correct the grammatical and spelling mistakes I found (too numerous to list).

Comments on the Quality of English Language

The review is written in excellent colloquial English.  However, the text needs careful proofreading to correct the grammatical and spelling mistakes I found (too numerous to list).

Author Response

Reviewer-1

  1. A Table summarizing the biochemical and biological properties of CCR5, including the cell types expressing it would be useful. 

Response: We agree with the reviewer that a table summarizing the properties of CCR5 would be helpful, and we also thought about including this in the original submission. Ultimately, we decided against this because there are numerous reviews and book chapters that discuss this in detail. In particular, there is an excellent review by Michael Lederman (PMID- 16905787) that summarizes the biochemical and biological properties of human CCR5, which we have now cited. As our review also focused on the role of mouse Ccr5, we thought to include the expression profile of Ccr5 on mouse cells, but because the mouse antibodies against Ccr5 are notoriously non-specific, we do not feel confident defining this currently. As mentioned in the review, new tools and reagents are needed to fully characterize this in mice.

  1. A Table summarizing the human and mouse studies of CCR5 for all the human flaviviruses would be useful.  

Response: We agree. We have now included this table.

Several flaviviruses are not mentioned in the review including yellow fever virus and dengue virus (PMID: 25939314, PMID: 24607451).  

Response: With regards to yellow fever and dengue virus, we chose to include only flaviviruses that are neurotropic because this review will be featured in a special issue focusing on neurotropic viruses. Neither yellow fever or dengue are classically defined as neurotropic, and thus we chose not to include these in our review.

Those mentioned at the end only should be worked into earlier sections of the review.  It’s important to convey that the role of CCR5 in flavivirus infection is not settled science as these discrepant studies make clear.

Response: With regards to the two flavivirus studies that show a pathogenic role of Ccr5, we completely agree with the reviewer. We have now created a new section (lines 364-381) in the manuscript to describe those studies where Ccr5 has been shown to promote pathogenesis. These are also now included in the new table we have created in response to the reviewers suggestion.

  1. The introduction assigns four words and one reference for the role of CCR5 in HIV infection.  Although the review is focused on flaviviruses, it is important to state that the role of CCR5 in viral infection is most clearly established for HIV, where it works as a direct entry factor/coreceptor with CD4 essential for R5 HIV infection of CD4+ T cells and macrophages.  And that CCR5delta32, which is central to many of the studies reviewed here, was discovered as part of a search to understand the essential role of CCR5 in AIDS pathogenesis.  Compared to this clearly established role with known mechanisms, understanding of CCR5’s apparently sub-essential role in flavivirus infection is still a work in progress

Response: We thank the reviewer for pointing this out and we agree entirely. In the revised review, we now include a more substantive section highlighting the role of CCR5 in HIV (lines 47-51).

  1. The lack of agreement among the gene association studies for WNV and CCR5D32 needs more comment regarding the limitation of such studies. Were any of these gold standard genome wide association studies?  Were any controlled for population structure?  Were they designed based on cohorts of convenience?  The characterization of the control populations is very important in interpreting these results.

Response: The reviewer is correct; we needed a more substantive section regarding this. We have now included more details about why these studies may be discrepant. Among the two studies that did not show an association with CCR5D32, one of them genotyped using PCR primers flanking the mutation site (similar to the original two studies showing an association), while the other relied on an Illumina based sequencing methodology. We do not know why these studies differ with regards to association with D32, but each of these studies were conducted differently with control populations that differ between them. Our studies utilized random blood donor controls, a reference cohort that was convenient, but were bona fide negative for WNV as they were collected prior to 1999. Similarly, utilization of an asymptomatic WNV-positive cohort (as done by Loeb et.al) is also not ideal as these are not a true negative population. We did our best to discuss these caveats in the revised manuscript (lines 157-167).

  1. Although the focus is on CCR5, there would certainly be interest among readers about other chemokines/chemokine receptors that have been shown to be as or more important in these infections.  The only one cited is a CXCL10 study.

Response: We agree with the reviewer. We have now included more information about what is known for each of these viruses with regards to chemokine/chemokine receptor regulation of pathogenesis (lines 80-88 as well as 180-188). Moreover, a detailed review of chemokines role in flavivirus infection has been published by our group where we discussed importance of various chemokines in flavivirus mediated pathogenesis (PMID- 22547394).

  1. The mechanistic aspects of CCR5’s role still appears to be weak.  I think there are floxed Ccr5 mice to define the role of CCR5 on specific leukocyte subtypes in flavivirus infection. 

Response: It appears that a floxed Ccr5 mouse is available at Shanghai Model Organism (https://www.modelorg.com/en/index.php/GEMs/32380/post_type/3.html). We contacted this organization a couple of years ago regarding these mice, but we were told that these mice are not validated. Further, there are no publications citing the use of the mice. Because of this, we do not know whether these mice are functioning as intended. We have also not seen any publications on the use of floxed Ccr5 mice in the context of flavivirus pathogenesis.

  1. The final reference is to a paper describing CCR5 reporter mice, but the precise cell types expressing CCR5 and related receptors described in the paper are not detailed in the review.  Readers will want to know what cell types express CCR5 ligands and CCR5 in the brain during flavivirus infection.  And whether CCR5 deficiency results in a defect in production of virus antigen-specific CD4 and/or CD8+ T cells.

Response: The reviewer is correct. The Medina-Ruiz paper did not define the expression of Ccr5 across hematopoietic and non-hematopoietic cells, but rather focuses on specific cell types and specific inflammatory contexts. We have now included the new information regarding Ccr5 expression that is provided by this reporter mouse in the manuscript (lines 419-420). 

Currently, there is no information about CCR5 expression on brain cells described at steady state or during flavivirus encephalitis. This is a major gap in the field as microglia and neurons may express Ccr5 in the CNS.

With regards to CCR5 deficiency leading to defects in mounting a T cell response, there are several papers that demonstrate that CCR5 is required for efficient activation and migration of CD8+ T cells, which utilize CCR5 to migrate to site of CD4:DC interactions (PMID: 16612374) or to sites of localized inflammation (PMID: 18617426) (lines 110-112).

  1. The published results with maraviroc are of questionable significance as the authors suggest.  Have any of the cited studies been performed with human CCR5 knockin mice?  

Response: To our knowledge, there are no studies using human CCR5 knockin mice and maraviroc.

What doses of the drug were used and are those doses able to achieve drug levels in the blood consistent with blocking mouse Ccr5?

We evaluated the dose of maraviroc used in the JEV study we cited and found that the dosing seems adequate based on our calculations (30mg/kg daily through oral route). Our back calculations based on the IC50 of Maraviroc on mouse Ccr5 shows that this is ~10x higher than what is required.  We have now addressed this in the review. We thank the reviewer for this comment (lines 398-399 as well as 403-405).

  1.   The WHO references need years and page numbers. 

Response: We obtained these numbers from the WHO website, rather than a publication. As such, we have replaced these citations with the URL.

All the references should be proofread since there appear to be grammatical errors in some of the titles. 

Response:  Since these were pulled directly from PubMed, they should not include any typos, but we have proofed these to doublecheck.

Also the text needs careful proofreading to correct the grammatical and spelling mistakes I found (too numerous to list).

Response:  We have also corrected the typos throughout the manuscripts.

Reviewer 2 Report

Comments and Suggestions for Authors

This review discusses the role of CCR5 in the pathophysiology of flavivirus infection and more particularly in the occurrence of neuromeningeal manifestations. The authors present the results of studies which have shown the epidemiological link between defect in CCR5 expression and severe forms of the infection in humans, and investigated the inflammatory mechanisms involved in mice. This reveals a neuroprotective role for CCR5 but according to sometimes opposing mechanisms ranging from the attraction of leukocytes in the central nervous system at the origin of viral clearance to a neuroprotective anti-inflammatory role. The article is clearly written and offers an exhaustive overview of the subject.

I only have a few very minor comments:

- The protective role of CCR5 is clearly demonstrated by studies carried out in mice while the epidemiological link observed in humans between CCR5Δ32 mutation and severity of infection is more vague with sometimes studies with contradictory results. The authors could argue this point.

- Even if we are on the periphery of the subject because the complications are more hemorrhagic, do we know the influence of the CCR5Δ32 mutation on the risk of occurrence of severe dengue?

- The authors will need to clarify lines 222-225 page 6 where two research groups are mentioned but only one bibliographic reference is cited. A few typos will also need to be corrected lines 228 (sufficient) and 229 (deficient).

Author Response

Reviewer-2

1-The protective role of CCR5 is clearly demonstrated by studies carried out in mice while the epidemiological link observed in humans between CCR5Δ32 mutation and severity of infection is more vague with sometimes studies with contradictory results. The authors could argue this point.

Response: We have rewritten the section discussing the epidemiological studies for WNV and CCR5D32 to add more information about these contradictory studies (lines 157-167). This was also a concern for Reviewer 1, which we have also addressed as written below:

     Among the two studies that did not show an association with CCR5D32, one of them genotyped using PCR primers flanking the mutation site (similar to the original two studies showing an association), while the other relied on an Illumina based sequencing methodology. We do not know why these studies differ with regards to association with D32, but each of these studies were conducted differently with control populations that differ between them. Our studies utilized random blood donor controls, a reference cohort that was convenient, but were bona fide negative for WNV as they were collected prior to 1999. Similarly, utilization of an asymptomatic WNV-positive cohort (as done by Loeb et.al) is also not ideal as these are not a true negative population. We did our best to discuss these caveats in the revised manuscript.

2- Even if we are on the periphery of the subject because the complications are more hemorrhagic, do we know the influence of the CCR5Δ32 mutation on the risk of occurrence of severe dengue?

Response: Thank you for this comment. Because our review focuses on neuroinvasive flaviviruses, we decided not to include dengue.  To address the reviewer’s question, there have been several studies evaluating dengue and CCR5D32, but all have been small studies (less than 100 dengue-infected individuals) and some in populations where the frequency of D32 homozygosity is expected to be extremely low. Mouse models are also largely not informative as dengue does not replicate in immunocompetent mice, and the models that do exist (AG129 mice) would not be informative regarding CCR5 due to the lack of an innate immune response.

3- The authors will need to clarify lines 222-225 page 6 where two research groups are mentioned but only one bibliographic reference is cited. A few typos will also need to be corrected lines 228 (sufficient) and 229 (deficient).

Response: We have fixed this error along with the typos in the manuscripts.

Round 2

Reviewer 1 Report

Comments and Suggestions for Authors

The revisions adequately address all my earlier comments.  

Comments on the Quality of English Language

There are a few grammatical mistakes in the added text.